# Generalizing Point Embeddings using the Wasserstein Space of Elliptical Distributions

**Boris Muzellec**
CREST, ENSAE
boris.muzellec@ensae.fr

**Marco Cuturi**
Google Brain and CREST, ENSAE
cuturi@google.com

## Abstract

Embedding complex objects as vectors in low dimensional spaces is a longstanding problem in machine learning. We propose in this work an extension of that approach, which consists in embedding objects as elliptical probability distributions, namely distributions whose densities have elliptical level sets. We endow these measures with the 2-Wasserstein metric, with two important benefits: *(i)* For such measures, the squared 2-Wasserstein metric has a closed form, equal to a weighted sum of the squared Euclidean distance between means and the squared Bures metric between covariance matrices. The latter is a Riemannian metric between positive semi-definite matrices, which turns out to be Euclidean on a suitable factor representation of such matrices, which is valid on the entire geodesic between these matrices. *(ii)* The 2-Wasserstein distance boils down to the usual Euclidean metric when comparing Diracs, and therefore provides a natural framework to extend point embeddings. We show that for these reasons Wasserstein elliptical embeddings are more intuitive and yield tools that are better behaved numerically than the alternative choice of Gaussian embeddings with the Kullback-Leibler divergence. In particular, and unlike previous work based on the KL geometry, we learn elliptical distributions that are not necessarily diagonal. We demonstrate the advantages of elliptical embeddings by using them for visualization, to compute embeddings of words, and to reflect entailment or hypernymy.

## 1 Introduction

One of the holy grails of machine learning is to compute meaningful low-dimensional embeddings for high-dimensional complex data. That ability has recently proved crucial to tackle more advanced tasks, such as for instance: inference on texts using word embeddings [Mikolov et al., 2013b, Pennington et al., 2014, Bojanowski et al., 2017], improved image understanding [Norouzi et al., 2014], representations for nodes in large graphs [Grover and Leskovec, 2016].

Such embeddings have been traditionally recovered by seeking *isometric* embeddings in lower dimensional Euclidean spaces, as studied in [Johnson and Lindenstrauss, 1984, Bourgain, 1985]. Given $n$ input points $x_1, \ldots, x_n$, one seeks as many embeddings $\mathbf{y}_1, \ldots, \mathbf{y}_n$ in a target space $\mathcal{Y} = \mathbb{R}^d$ whose pairwise distances $\|\mathbf{y}_i - \mathbf{y}_j\|_2$ do not depart too much from the original distances $d_{\mathcal{X}}(x_i, x_j)$ in the input space. Note that when $d$ is restricted to be 2 or 3, these embeddings $(\mathbf{y}_i)_i$ provide a useful way to visualize the entire dataset. Starting with metric multidimensional scaling (mMDS) [De Leeuw, 1977, Borg and Groenen, 2005], several approaches have refined this intuition [Tenenbaum et al., 2000, Roweis and Saul, 2000, Hinton and Roweis, 2003, Maaten and Hinton, 2008]. More general criteria, such as reconstruction error [Hinton and Salakhutdinov, 2006, Kingma and Welling, 2014]; co-occurence [Globerson et al., 2007]; or relational knowledge, be it in metric learning [Weinberger and Saul, 2009] or between words [Mikolov et al., 2013b] can be used to obtain vector embeddings. In such cases, distances $\|\mathbf{y}_i - \mathbf{y}_j\|_2$ between embeddings, or alternatively their dot-products $\langle \mathbf{y}_i, \mathbf{y}_j \rangle$

must comply with sophisticated desiderata. Naturally, more general and flexible approaches in which the embedding space $\mathcal{Y}$ needs not be Euclidean can be considered, for instance in generalized MDS on the sphere [Maron et al., 2010], on surfaces [Bronstein et al., 2006], in spaces of trees [Bădoiu et al., 2007, Fakcharoenphol et al., 2003] or, more recently, computed in the Poincaré hyperbolic space [Nickel and Kiela, 2017].

**Probabilistic Embeddings.** Our work belongs to a recent trend, pioneered by Vilnis and McCallum, who proposed to embed data points as *probability measures* in $\mathbb{R}^d$ [2015], and therefore generalize point embeddings. Indeed, point embeddings can be regarded as a very particular—and degenerate—case of probabilistic embedding, in which the uncertainty is infinitely concentrated on a single point (a Dirac). Probability measures can be more spread-out, or event multimodal, and provide therefore an opportunity for additional flexibility. Naturally, such an opportunity can only be exploited by defining a metric, divergence or dot-product on the space (or a subspace thereof) of probability measures. Vilnis and McCallum proposed to embed words as *Gaussians* endowed either with the Kullback-Leibler (KL) divergence or the expected likelihood kernel [Jebara et al., 2004]. The Kullback-Leibler and expected likelihood kernel on measures have, however, an important drawback: these geometries do not coincide with the usual Euclidean metric between point embeddings when the variances of these Gaussians collapse. Indeed, the KL divergence and the $\ell_2$ distance between two Gaussians diverges to $\infty$ or saturates when the variances of these Gaussians become small. To avoid numerical instabilities arising from this degeneracy, Vilnis and McCallum must restrict their work to diagonal covariance matrices. In a concurrent approach, Singh et al. represent words as distributions over their contexts in the optimal transport geometry [Singh et al., 2018].

**Contributions.** We propose in this work a new framework for probabilistic embeddings, in which point embeddings are seamlessly handled as a particular case. We consider arbitrary families of elliptical distributions, which subsume Gaussians, and also include uniform elliptical distributions, which are arguably easier to visualize because of their compact support. Our approach uses the 2-Wasserstein distance to compare elliptical distributions. The latter can handle degenerate measures, and both its value and its gradients admit closed forms [Gelbrich, 1990], either in their natural Riemannian formulation, as well as in a more amenable local Euclidean parameterization. We provide numerical tools to carry out the computation of elliptical embeddings in different scenarios, both to optimize them with respect to metric requirements (as is done in multidimensional scaling) or with respect to dot-products (as shown in our applications to word embeddings for entailment, similarity and hypernymy tasks) for which we introduce a proxy using a polarization identity.

**Notations** $\mathcal{S}_{++}^d$ (resp. $\mathcal{S}_+^d$) is the set of positive (resp. semi-)definite $d \times d$ matrices. For two vectors $\mathbf{x}, \mathbf{y} \in \mathbb{R}^d$ and a matrix $\mathbf{M} \in \mathcal{S}_+^d$, we write the Mahalanobis norm induced by $\mathbf{M}$ as $\|\mathbf{x} - \mathbf{c}\|_{\mathbf{M}}^2 = (\mathbf{x} - \mathbf{c})^T \mathbf{M} (\mathbf{x} - \mathbf{c})$ and $|\mathbf{M}|$ for $\det(\mathbf{M})$. For $V$ an affine subspace of dimension $m$ of $\mathbb{R}^d$, $\lambda_V$ is the Lebesgue measure on that subspace. $\mathbf{M}^\dagger$ is the pseudo inverse of $\mathbf{M}$.

## 2 The Geometry of Elliptical Distributions in the Wasserstein Space

We recall in this section basic facts about elliptical distributions in $\mathbb{R}^d$. We adopt a general formulation that can handle measures supported on subspaces of $\mathbb{R}^d$ as well as Dirac (point) measures. That level of generality is needed to provide a seamless connection with usual vector embeddings, seen in the context of this paper as Dirac masses. We recall results from the literature showing that the squared 2-Wasserstein distance between two distributions from the same family of elliptical distributions is equal to the squared Euclidean distance between their means plus the squared Bures metric between their scale parameter scaled by a suitable constant.

**Elliptically Contoured Densities.** In their simplest form, elliptical distributions can be seen as generalizations of Gaussian multivariate densities in $\mathbb{R}^d$: their level sets describe concentric ellipsoids, shaped following a scale parameter $\mathbf{C} \in \mathcal{S}_{++}^d$, and centered around a mean parameter $\mathbf{c} \in \mathbb{R}^d$ [Cambanis et al., 1981]. The density at a point $\mathbf{x}$ of such distributions is $f(\|\mathbf{x} - \mathbf{c}\|_{\mathbf{C}^{-1}})/\sqrt{|\mathbf{C}|}$ where the generator function $f$ is such that $\int_{\mathbb{R}^d} f(\|\mathbf{x}\|^2) d\mathbf{x} = 1$. Gaussians are recovered with $f = g, g(\cdot) \propto e^{-\cdot/2}$ while uniform distributions on full rank ellipsoids result from $f = u, u(\cdot) \propto \mathbf{1}_{\cdot \leq 1}$.

Because the norm induced by $\mathbf{C}^{-1}$ appears in formulas above, the scale parameter $\mathbf{C}$ must have full rank for these definitions to be meaningful. Cases where $\mathbf{C}$ does not have full rank can however

appear when a probability measure is supported on an affine subspace[1] of $\mathbb{R}^d$, such as lines in $\mathbb{R}^2$, or even possibly a space of null dimension when the measure is supported on a single point (a Dirac measure), in which case its scale parameter $\mathbf{C}$ is $\mathbf{0}$. We provide in what follows a more general approach to handle these degenerate cases.

**Elliptical Distributions.** To lift this limitation, several reformulations of elliptical distributions have been proposed to handle degenerate scale matrices $\mathbf{C}$ of rank $\operatorname{rk}\mathbf{C} < d$. Gelbrich [1990, Theorem 2.4] defines elliptical distributions as measures with a density w.r.t the Lebesgue measure of dimension $\operatorname{rk}\mathbf{C}$, in the affine space $\mathbf{c} + \operatorname{Im}\mathbf{C}$, where the image of $\mathbf{C}$ is $\operatorname{Im}\mathbf{C} \stackrel{\text{def}}{=} \{\mathbf{C}\mathbf{x}, \mathbf{x} \in \mathbb{R}^d\}$. This approach is intuitive, in that it reduces to describing densities in their relevant subspace. A more elegant approach uses the parameterization provided by characteristic functions [Cambanis et al., 1981, Fang et al., 1990]. In a nutshell, recall that the characteristic function of a multivariate Gaussian is equal to $\phi(\mathbf{t}) = e^{i\mathbf{t}^T\mathbf{c}}g(\mathbf{t}^T\mathbf{C}\mathbf{t})$ where, as in the paragraph above, $g(\cdot) = e^{-\cdot/2}$. A natural generalization to consider other elliptical distributions is therefore to consider for $g$ other functions $h$ of positive type [Ushakov, 1999, Theo.1.8.9], such as the indicator function $u$ above, and still apply them to the same argument $\mathbf{t}^T\mathbf{C}\mathbf{t}$. Such functions are called *characteristic generators* and fully determine, along with a mean $\mathbf{c}$ and a scale parameter $\mathbf{C}$, an elliptical measure. This parameterization does not require the scale parameter $\mathbf{C}$ to be invertible, and therefore allows to define probability distributions that do not have necessarily a density w.r.t to the Lebesgue measure in $\mathbb{R}^d$. Both constructions are relatively complex, and we refer the interested reader to these references for a rigorous treatment.

**Rank Deficient Elliptical Distributions and their Variances.** For the purpose of this work, we will only require the following result: the variance of an elliptical measure is equal to its scale parameter $\mathbf{C}$ multiplied by a scalar that only depends on its characteristic generator. Indeed, given a mean vector $\mathbf{c} \in \mathbb{R}^d$, a scale *semi*-definite matrix $\mathbf{C} \in \mathcal{S}_+^d$ and a characteristic generator function $h$, we define $\mu_{h,\mathbf{c},\mathbf{C}}$ to be the measure with characteristic function $\mathbf{t} \mapsto e^{i\mathbf{t}^T\mathbf{c}}h(\mathbf{t}^T\mathbf{C}\mathbf{t})$. In that case, one can show that the covariance matrix of $\mu_{h,\mathbf{c},\mathbf{C}}$ is equal to its scale parameter $\mathbf{C}$ times a constant $\tau_h$ that only depends on $h$, namely

$$\operatorname{var}(\mu_{h,\mathbf{c},\mathbf{C}}) = \tau_h\mathbf{C} \ . \tag{1}$$

For Gaussians, the scale parameter $\mathbf{C}$ and its covariance matrice coincide, that is $\tau_g = 1$. For uniform elliptical distributions, one has $\tau_u = 1/(d+2)$: the covariance of a uniform distribution on the volume $\{\mathbf{c}+\mathbf{C}\mathbf{x}, \mathbf{x} \in \mathbb{R}^d, \|\mathbf{x}\| = 1\}$, such as those represented in Figure 1, is equal to $\mathbf{C}/(d+2)$.

**The 2-Wasserstein Bures Metric** A natural metric for elliptical distributions arises from optimal transport (OT) theory. We refer interested readers to [Santambrogio, 2015, Peyré and Cuturi, 2018] for exhaustive surveys on OT. Recall that for two arbitrary probability measures $\mu, \nu \in \mathcal{P}(\mathbb{R}^d)$, their squared 2-Wasserstein distance is equal to

$$W_2^2(\mu,\nu) \stackrel{\text{def}}{=} \inf_{X \sim \mu, Y \sim \nu} \mathbb{E}_{\|X-Y\|_2^2}.$$

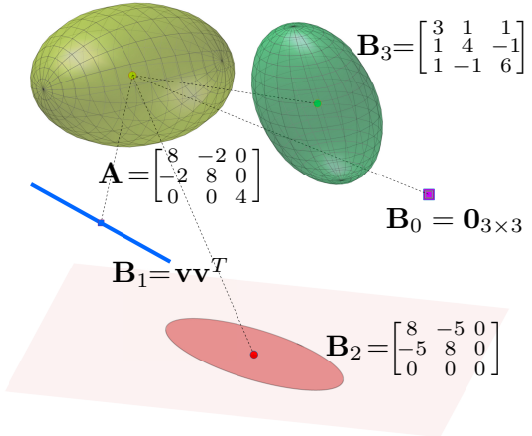

Figure 1: Five measures from the family of uniform elliptical distributions in $\mathbb{R}^3$. Each measure has a mean (location) and scale parameter. In this carefully selected example, the reference measure (with scale parameter $\mathbf{A}$) is equidistant (according to the 2-Wasserstein metric) to the four remaining measures, whose scale parameters $\mathbf{B}_0, \mathbf{B}_1, \mathbf{B}_2, \mathbf{B}_3$ have ranks equal to their indices (here, $\mathbf{v} = [3, 7, -2]^T$).

This formula rarely has a closed form. However, in the footsteps of Dowson and Landau [1982] who proved it for Gaussians, Gelbrich [1990] showed that for $\alpha \stackrel{\text{def}}{=} \mu_{h,\mathbf{a},\mathbf{A}}$ and $\beta \stackrel{\text{def}}{=} \mu_{h,\mathbf{b},\mathbf{B}}$ in the *same* family $\mathcal{P}_h = \{\mu_{h,\mathbf{c},\mathbf{C}}, \mathbf{c} \in \mathbb{R}^d, \mathbf{C} \in \mathcal{S}_+^d\}$, one has

$$W_2^2(\alpha,\beta) = \|\mathbf{a} - \mathbf{b}\|_2^2 + \mathfrak{B}^2(\operatorname{var}\alpha, \operatorname{var}\beta) = \|\mathbf{a} - \mathbf{b}\|_2^2 + \tau_h\mathfrak{B}^2(\mathbf{A}, \mathbf{B}) \ , \tag{2}$$

where $\mathfrak{B}^2$ is the (squared) Bures metric on $\mathcal{S}_+^d$, proposed in quantum information geometry [1969] and studied recently in [Bhatia et al., 2018, Malagò et al., 2018],

$$\mathfrak{B}^2(\mathbf{X}, \mathbf{Y}) \stackrel{\text{def}}{=} \text{Tr}(\mathbf{X} + \mathbf{Y} - 2(\mathbf{X}^{\frac{1}{2}} \mathbf{Y} \mathbf{X}^{\frac{1}{2}})^{\frac{1}{2}}) \ . \tag{3}$$

The factor $\tau_h$ next to the rightmost term $\mathfrak{B}^2$ in (2) arises from homogeneity of $\mathfrak{B}^2$ in its arguments (3), which is leveraged using the identity in (1).

**A few remarks** *(i)* When both scale matrices $\mathbf{A} = \text{diag}\,\mathbf{d_A}$ and $\mathbf{B} = \text{diag}\,\mathbf{d_B}$ are diagonal, $W_2^2(\alpha, \beta)$ is the sum of two terms: the usual squared Euclidean distance between their means, plus $\tau_h$ times the squared *Hellinger* metric between the diagonals $\mathbf{d_A}, \mathbf{d_B}$: $\mathfrak{H}^2(\mathbf{d_A}, \mathbf{d_B}) \stackrel{\text{def}}{=} \|\sqrt{\mathbf{d_A}} - \sqrt{\mathbf{d_B}}\|_2^2$. *(ii)* The distance $W_2$ between two Diracs $\delta_\mathbf{a}, \delta_\mathbf{b}$ is equal to the usual distance between vectors $\|\mathbf{a} - \mathbf{b}\|_2$. *(iii)* The squared distance $W_2^2$ between a Dirac $\delta_\mathbf{a}$ and a measure $\mu_{h,\mathbf{b},\mathbf{B}}$ in $\mathcal{P}_h$ reduces to $\|\mathbf{a} - \mathbf{b}\|^2 + \tau_h \text{Tr}\mathbf{B}$. The distance between a point and an ellipsoid distribution therefore always *increases* as the scale parameter of the latter increases. Although this point makes sense from the quadratic viewpoint of $W_2^2$ (in which the quadratic contribution $\|\mathbf{a} - \mathbf{x}\|_2^2$ of points $\mathbf{x}$ in the ellipsoid that stand further away from $\mathbf{a}$ than $\mathbf{b}$ will dominate that brought by points $\mathbf{x}$ that are closer, see Figure 3) this may be counterintuitive for applications to visualization, an issue that will be addressed in Section 4. *(iv)* The $W_2$ distance between two elliptical distributions in the same family $\mathcal{P}_h$ is always finite, no matter how degenerate they are. This is illustrated in Figure 1 in which a uniform measure $\mu_{\mathbf{a},\mathbf{A}}$ is shown to be exactly equidistant to four other uniform elliptical measures, some of which are degenerate. However, as can be hinted by the simple example of the Hellinger metric, that distance may not be differentiable for degenerate measures (in the same sense that $(\sqrt{x} - \sqrt{y})^2$ is defined at $x = 0$ but not differentiable w.r.t $x$). *(v)* Although we focus in this paper on uniform elliptical distributions, notably because they are easier to plot and visualize, considering any other elliptical family simply amounts to changing the constant $\tau_h$ next to the Bures metric in (2). Alternatively, increasing (or tuning) that parameter $\tau_h$ simply amounts to considering elliptical distributions with increasingly heavier tails.

## 3 Optimizing over the Space of Elliptical Embeddings

Our goal in this paper is to use the set of elliptical distributions endowed with the $W_2$ distance as an embedding space. To optimize objective functions involving $W_2$ terms, we study in this section several parameterizations of the parameters of elliptical distributions. Location parameters only appear in the computation of $W_2$ through their Euclidean metric, and offer therefore no particular challenge. Scale parameters are more tricky to handle since they are constrained to lie in $\mathcal{S}_+^d$. Rather than keeping track of scale parameters, we advocate optimizing directly on factors (square roots) of such parameters, which results in simple Euclidean (unconstrained) updates reviewed below.

**Geodesics for Elliptical Distributions** When $\mathbf{A}$ and $\mathbf{B}$ have full rank, the geodesic from $\alpha$ to $\beta$ is a curve of measures in the same family of elliptic distributions, characterized by location and scale parameters $\mathbf{c}(t), \mathbf{C}(t)$, where

$$\mathbf{c}(t) = (1-t)\mathbf{a} + t\mathbf{b}; \quad \mathbf{C}(t) = \left((1-t)\mathbf{I} + t\mathbf{T^{AB}}\right)\mathbf{A}\left((1-t)\mathbf{I} + t\mathbf{T^{AB}}\right) \ , \tag{4}$$

and where the matrix $\mathbf{T^{AB}}$ is such that $\mathbf{x} \to \mathbf{T^{AB}}(\mathbf{x} - \mathbf{a}) + \mathbf{b}$ is the so-called Brenier optimal transportation map [1987] from $\alpha$ to $\beta$, given in closed form as,

$$\mathbf{T^{AB}} \stackrel{\text{def}}{=} \mathbf{A}^{-\frac{1}{2}}(\mathbf{A}^{\frac{1}{2}}\mathbf{B}\mathbf{A}^{\frac{1}{2}})^{\frac{1}{2}}\mathbf{A}^{-\frac{1}{2}} \ , \tag{5}$$

and is the unique matrix such that $\mathbf{B} = \mathbf{T^{AB}}\mathbf{A}\mathbf{T^{AB}}$ [Peyré and Cuturi, 2018, Remark 2.30]. When $\mathbf{A}$ is degenerate, such a curve still exists as long as $\text{Im}\,\mathbf{B} \subset \text{Im}\,\mathbf{A}$, in which case the expression above is still valid using pseudo-inverse square roots $\mathbf{A}^{\dagger/2}$ in place of the usual inverse square-root.

**Differentiability in Riemannian Parameterization** Scale parameters are restricted to lie on the cone $\mathcal{S}_+^d$. For such problems, it is well known that a direct gradient-and-project based optimization on scale parameters would prove too expensive. A natural remedy to this issue is to perform manifold optimization [Absil et al., 2009]. Indeed, as in any Riemannian manifold, the Riemannian gradient $\text{grad}_x \frac{1}{2} d^2(x, y)$ is given by $-\log_x y$ [Lee, 1997]. Using the expressions of the $\exp$ and $\log$ given in [Malagò et al., 2018], we can show that minimizing $\frac{1}{2}\mathfrak{B}^2(\mathbf{A}, \mathbf{B})$ using Riemannian gradient descent corresponds to making updates of the form, with step length $\eta$

$$\mathbf{A}' = \left((1-\eta)\mathbf{I} + \eta\mathbf{T^{AB}}\right)\mathbf{A}\left((1-\eta)\mathbf{I} + \eta\mathbf{T^{AB}}\right) \ . \tag{6}$$

When $0 \leq \eta \leq 1$, this corresponds to considering a new point $\mathbf{A}'$ closer to $\mathbf{B}$ along the Bures geodesic between $\mathbf{A}$ and $\mathbf{B}$. When $\eta$ is negative or larger than 1, $\mathbf{A}'$ no longer lies on this geodesic but is guaranteed to remain PSD, as can be seen from (6). Figure 2 shows a $W_2$ geodesic between two measures $\mu_0$ and $\mu_1$, as well as its extrapolation following exactly the formula given in (4). That figure illustrates that $\mu_t$ is not necessarily geodesic outside of the boundaries $[0, 1]$ w.r.t. three relevant measures, because its metric derivative is smaller than 1 [Ambrosio et al., 2006, Theorem 1.1.2]. When negative steps are taken (for instance when the $W_2^2$ distance needs to be increased), this lack of geodisicity has proved difficult to handle numerically for a simple reason: such updates may lead to degenerate scale parameters $\mathbf{A}'$, as illustrated around time $t = 1.5$ of the curve in Figure 2. Another obvious drawback of Riemannian approaches is that they are not as well studied as simpler non-constrained Euclidean problems, for which a plethora of optimization techniques are available. This observations motivates an alternative Euclidean parameterization, detailed in the next paragraph.

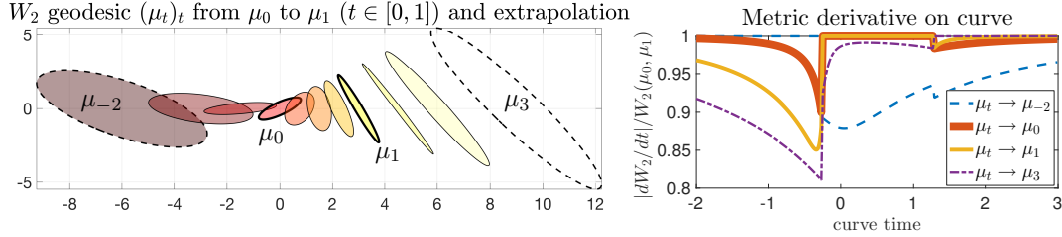

Figure 2: (left) Interpolation $(\mu_t)_t$ between two measures $\mu_0$ and $\mu_1$ following the geodesic equation (4). The same formula can be used to interpolate on the left and right of times $0, 1$. Displayed times are $[-2, -1, -.5, 0, .25, .5, .75, 1, 1.5, 2, 3]$. Note that geodesicity is not ensured outside of the boundaries $[0, 1]$. This is illustrated in the right plot displaying normalized metric derivatives of the curve $\mu_t$ to four relevant points: $\mu_0, \mu_1, \mu_{-2}, \mu_3$. The curve $\mu_t$ is not always locally geodesic, as can be seen by the fact that the metric derivative is strictly smaller than 1 in several cases.

**Differentiability in Euclidean Parameterization** A canonical way to handle a PSD constraint for $\mathbf{A}$ is to rewrite it in factor form $\mathbf{A} = \mathbf{LL}^T$. In the particular case of the Bures metric, we show that this simple parametrization comes without losing the geometric interest of manifold optimization, while benefiting from simpler additive updates. Indeed, one can (see supplementary material) that the gradient of the squared Bures metric has the following gradient:

$$\nabla_{\mathbf{L}} \frac{1}{2} \mathfrak{B}^2(\mathbf{A}, \mathbf{B}) = (\mathbf{I} - \mathbf{T^{AB}}) \mathbf{L}, \quad \text{with updates } \mathbf{L}' = ((1 - \eta)\mathbf{I} + \eta\mathbf{T^{AB}}) \mathbf{L} \ . \quad (7)$$

**Links between Euclidean and Riemannian Parameterization** The factor updates in (7) are exactly equivalent to the Riemannian ones (6) in the sense that $\mathbf{A}' = \mathbf{L}'\mathbf{L}'^T$. Therefore, by using a factor parameterization we carry out updates that stay on the Riemannian geodesic yet only require linear updates on $\mathbf{L}$, independently of the factor $\mathbf{L}$ chosen to represent $\mathbf{A}$ (given a factor $\mathbf{L}$ of $\mathbf{A}$, any right-side multiplication of that matrix by a unitary matrix remains a factor of $\mathbf{A}$).

When considering a general loss function $\mathcal{L}$ that take as arguments squared Bures distances, one can also show that $\mathcal{L}$ is geodesically convex w.r.t. to scale matrices $\mathbf{A}$ if and only if it is convex in the usual sense with respect to $\mathbf{L}$, where $\mathbf{A} = \mathbf{LL}^T$. Write now $\mathbf{L_B} = \mathbf{T^{AB}L}$. One can recover that $\mathbf{L_B}\mathbf{L_B}^T = \mathbf{B}$. Therefore, expanding the expression $\mathfrak{B}^2$ for the right term below we obtain

$$\mathfrak{B}^2(\mathbf{A}, \mathbf{B}) = \mathfrak{B}^2 (\mathbf{LL}^T, \mathbf{L_B}\mathbf{L_B}^T) = \mathfrak{B}^2 (\mathbf{LL}^T, \mathbf{T^{AB}L} (\mathbf{T^{AB}L})^T) = \|\mathbf{L} - \mathbf{T^{AB}L}\|_F^2$$

Indeed, the Bures distance simply reduces to the Frobenius distance between two factors of $\mathbf{A}$ and $\mathbf{B}$. However these factors need to be carefully chosen: given $\mathbf{L}$ for $\mathbf{A}$, the factor for $\mathbf{B}$ must be computed according to an optimal transport map $\mathbf{T^{AB}}$.

**Polarization between Elliptical Distributions** Some of the applications we consider, such as the estimation of word embeddings, are inherently based on dot-products. By analogy with the polarization identity, $\langle \mathbf{x}, \mathbf{y} \rangle = (\|\mathbf{x} - \mathbf{0}\|^2 + \|\mathbf{y} - \mathbf{0}\|^2 - \|\mathbf{x} - \mathbf{y}\|^2)/2$, we define a Wasserstein-Bures *pseudo*-dot-product, where $\delta_{\mathbf{0}} = \mu_{\mathbf{0}_d, \mathbf{0}_{d \times d}}$ is the Dirac mass at $\mathbf{0}$,

$$[\mu_{\mathbf{a},\mathbf{A}} : \mu_{\mathbf{b},\mathbf{B}}] \stackrel{\text{def}}{=} \frac{1}{2} \left( W_2^2(\mu_{\mathbf{a},\mathbf{A}}, \delta_{\mathbf{0}}) + W_2^2(\mu_{\mathbf{b},\mathbf{B}}, \delta_{\mathbf{0}}) - W_2^2(\mu_{\mathbf{a},\mathbf{A}}, \mu_{\mathbf{b},\mathbf{B}}) \right) = \langle \mathbf{a}, \mathbf{b} \rangle + \mathrm{Tr} \, (\mathbf{A}^{\frac{1}{2}}\mathbf{B}\mathbf{A}^{\frac{1}{2}})^{\frac{1}{2}}$$

Note that $[\cdot : \cdot]$ is not an actual inner product since the Bures metric is not Hilbertian, unless we restrict ourselves to diagonal covariance matrices, in which case it is the the inner product between $(\mathbf{a}, \sqrt{\mathbf{d_A}})$ and $(\mathbf{b}, \sqrt{\mathbf{d_B}})$. We use $[\mu_{\mathbf{a},\mathbf{A}} : \mu_{\mathbf{b},\mathbf{B}}]$ as a similarity measure which has, however, some regularity: one can show that when $\mathbf{a}, \mathbf{b}$ are constrained to have equal norms and $\mathbf{A}$ and $\mathbf{B}$ equal traces, then $[\mu_{\mathbf{a},\mathbf{A}} : \mu_{\mathbf{b},\mathbf{B}}]$ is maximal when $\mathbf{a} = \mathbf{b}$ and $\mathbf{A} = \mathbf{B}$. Differentiating all three terms in that sum, the gradient of this pseudo dot-product w.r.t. $\mathbf{A}$ reduces to $\nabla_{\mathbf{A}}[\mu_{\mathbf{a},\mathbf{A}} : \mu_{\mathbf{b},\mathbf{B}}] = \mathbf{T}^{\mathbf{AB}}$.

**Computational Aspects** The computational bottleneck of gradient-based Bures optimization lies in the matrix square roots and inverse square roots operations that arise when instantiating transport maps $\mathbf{T}$ as in (5). A naive method using eigenvector decomposition is far too time-consuming, and there is not yet, to the best of our knowledge, a straightforward way to perform it in batches on a GPU. We propose to use Newton-Schulz iterations (Algorithm 1, see [Higham, 2008, Ch. 6]) to approximate these root computations. These iterations producing both a root and an inverse root approximation, and, relying exclusively on matrix-matrix multiplications, stream efficiently on GPUs. Another problem lies in the fact that numerous roots and inverse-roots are required to form map $\mathbf{T}$. To solve this, we exploit an alternative formula for $\mathbf{T}^{\mathbf{AB}}$ (proof in the supplementary material):

$$\mathbf{T}^{\mathbf{AB}} = \mathbf{A}^{-\frac{1}{2}}(\mathbf{A}^{\frac{1}{2}}\mathbf{B}\mathbf{A}^{\frac{1}{2}})^{\frac{1}{2}}\mathbf{A}^{-\frac{1}{2}} = \mathbf{B}^{\frac{1}{2}}(\mathbf{B}^{\frac{1}{2}}\mathbf{A}\mathbf{B}^{\frac{1}{2}})^{-\frac{1}{2}}\mathbf{B}^{\frac{1}{2}}. \tag{8}$$

In a gradient update, both the loss and the gradient of the metric are needed. In our case, we can use the matrix roots computed during loss evaluation and leverage the identity above to compute on a budget the gradients with respect to either scale matrices $\mathbf{A}$ and $\mathbf{B}$. Indeed, a naive computation of $\nabla_{\mathbf{A}}\mathfrak{B}^2(\mathbf{A}, \mathbf{B})$ and $\nabla_{\mathbf{B}}\mathfrak{B}^2(\mathbf{A}, \mathbf{B})$ would require the knowledge of 6 roots:

$$\mathbf{A}^{\frac{1}{2}}, \mathbf{B}^{\frac{1}{2}}, (\mathbf{A}^{\frac{1}{2}}\mathbf{B}\mathbf{A}^{\frac{1}{2}})^{\frac{1}{2}}, (\mathbf{B}^{\frac{1}{2}}\mathbf{A}\mathbf{B}^{\frac{1}{2}})^{\frac{1}{2}}, \mathbf{A}^{-\frac{1}{2}}, \text{ and } \mathbf{B}^{-\frac{1}{2}}$$

to compute the following transport maps

$$\mathbf{T}^{\mathbf{AB}} = \mathbf{A}^{-\frac{1}{2}}(\mathbf{A}^{\frac{1}{2}}\mathbf{B}\mathbf{A}^{\frac{1}{2}})^{\frac{1}{2}}\mathbf{A}^{-\frac{1}{2}}, \mathbf{T}^{\mathbf{BA}} = \mathbf{B}^{-\frac{1}{2}}(\mathbf{B}^{\frac{1}{2}}\mathbf{A}\mathbf{B}^{\frac{1}{2}})^{\frac{1}{2}}\mathbf{B}^{-\frac{1}{2}} ,$$

namely four matrix roots and two matrix inverse roots. We can avoid computing those six matrices using identity (8) and limit ourselves to two runs of Algorithm 1, to obtain the same quantities as

$$\{\mathbf{Y}_1 \overset{\text{def}}{=} \mathbf{A}^{\frac{1}{2}}, \mathbf{Z}_1 \overset{\text{def}}{=} \mathbf{A}^{-\frac{1}{2}}\}, \{\mathbf{Y}_2 \overset{\text{def}}{=} (\mathbf{A}^{\frac{1}{2}}\mathbf{B}\mathbf{A}^{\frac{1}{2}})^{\frac{1}{2}}, \mathbf{Z}_2 \overset{\text{def}}{=} (\mathbf{A}^{\frac{1}{2}}\mathbf{B}\mathbf{A}^{\frac{1}{2}})^{-\frac{1}{2}}\}$$
$$\mathbf{T}^{\mathbf{AB}} = \mathbf{Z}_1\mathbf{Y}_2\mathbf{Z}_1, \mathbf{T}^{\mathbf{BA}} = \mathbf{Y}_1\mathbf{Z}_2\mathbf{Y}_1 .$$

When computing the gradients of $n \times m$ squared Wasserstein distances $W_2^2(\alpha_i, \beta_j)$ in parallel, one only needs to run $n$ Newton-Schulz algorithms (in parallel) to compute matrices $(\mathbf{Y}_1^i, \mathbf{Z}_1^i)_{i \leq n}$, and then $n \times m$ Newton-Schulz algorithms to recover cross matrices $\mathbf{Y}_2^{i,j}, \mathbf{Z}_2^{i,j}$. On the other hand, using an automatic differentiation framework would require an additional backward computation of the same complexity as the forward pass evaluating computation of the roots and inverse roots, hence requiring roughly twice as many operations per batch.

**Avoiding Rank Deficiency at Optimization Time** Although $\mathfrak{B}^2(\mathbf{A}, \mathbf{B})$ is defined for rank deficient matrices $\mathbf{A}$ and $\mathbf{B}$, it is not differentiable with respect to these matrices if they are rank deficient. Indeed, as mentioned earlier, this can be compared to the non-differentiability of the Hellinger metric, $(\sqrt{x} - \sqrt{y})^2$ when $x$

---

**Algorithm 1** Newton-Schulz

**Input:** PSD matrix $\mathbf{A}$, $\epsilon > 0$
  $\mathbf{Y} \leftarrow \frac{\mathbf{A}}{(1+\epsilon)\|\mathbf{A}\|}, \mathbf{Z} \leftarrow \mathbf{I}$
  **while** not converged **do**
    $\mathbf{T} \leftarrow (3\mathbf{I} - \mathbf{Z}\mathbf{Y})/2$
    $\mathbf{Y} \leftarrow \mathbf{Y}\mathbf{T}$
    $\mathbf{Z} \leftarrow \mathbf{T}\mathbf{Z}$
  **end while**
  $\mathbf{Y} \leftarrow \sqrt{(1+\epsilon)\|\mathbf{A}\|}\mathbf{Y}$
  $\mathbf{Z} \leftarrow \frac{\mathbf{Z}}{\sqrt{(1+\epsilon)\|\mathbf{A}\|}}$
**Output:** square root $\mathbf{Y}$, inverse square root $\mathbf{Z}$

---

or $y$ becomes $0$, at which point if becomes *not* differentiable. If $\operatorname{Im}\mathbf{B} \not\subset \operatorname{Im}\mathbf{A}$, which is notably the case if $\operatorname{rk}\mathbf{B} > \operatorname{rk}\mathbf{A}$, then $\nabla_{\mathbf{A}}\mathfrak{B}^2(\mathbf{A}, \mathbf{B})$ no longer exists. However, even in that case, $\nabla_{\mathbf{B}}\mathfrak{B}^2(\mathbf{A}, \mathbf{B})$ exists iff $\operatorname{Im}\mathbf{A} \subset \operatorname{Im}\mathbf{B}$. Since it would be cumbersome to account for these subtleties in a large scale optimization setting, we propose to add a small common regularization term to all the factor products considered for our embeddings, and set $\mathbf{A}_\varepsilon = \mathbf{L}\mathbf{L}^T + \varepsilon\mathbf{I}$ were $\varepsilon > 0$ is a hyperparameter. This ensures that all matrices are full rank, and thus that all gradients exist. Most importantly, all our derivations still hold with this regularization, and can be shown to leave the method to compute the gradients w.r.t $\mathbf{L}$ unchanged, namely remain equal to $(\mathbf{I} - \mathbf{T}^{\mathbf{A}_\varepsilon\mathbf{B}})\mathbf{L}$.

# 4 Experiments

We discuss in this section several applications of elliptical embeddings. We first consider a simple mMDS type visualization task, in which elliptical distributions in $d = 2$ are used to embed isometrically points in high dimension. We argue that for such purposes, a more natural way to visualize ellipses is to use their precision matrices. This is due to the fact that the human eye somewhat acts in the opposite direction to the Bures metric, as discussed in Figure 3. We follow with more advanced experiments in which we consider the task of computing word embeddings on large corpora as a testing ground, and equal or improve on the state-of-the-art.

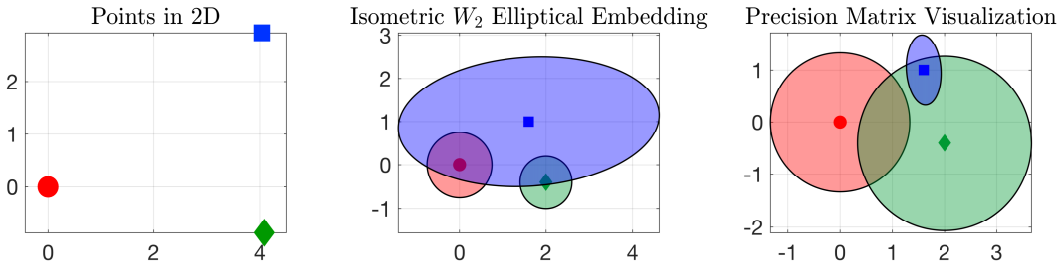

Figure 3: (left) three points on the plane. (middle) *isometric* elliptic embedding with the Bures metric: ellipses of a given color have the same respective distances as points on the left. Although the mechanics of optimal transport indicate that the blue ellipsoid is far from the two others, in agreement with the left plot, the human eye tends to focus on those areas that overlap (below the ellipsoid center) rather than those far away areas (north-east area) that contribute more significantly to the $W_2$ distance. (right) the precision matrix visualization, obtained by considering ellipses with the same axes but inverted eigenvalues, agree better with intuition, since they emphasize that overlap and extension of the ellipse means on the contrary that those axis contribute less to the increase of the metric.

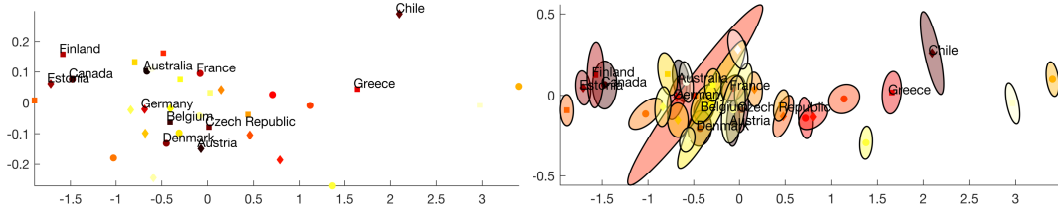

Figure 4: Toy experiment: visualization of a dataset of 10 PISA scores for 35 countries in the OECD. (left) MDS embeddings of these countries on the plane (right) elliptical embeddings on the plane using the precision visualization discussed in Figure 3. The normalized stress with standard MDS is 0.62. The stress with elliptical embeddings is close to $5e - 3$ after 1000 gradient iterations, with random initializations for scale matrices (following a Standard Wishart with 4 degrees of freedom) and initial means located on the MDS solution.

**Visualizing Datasets Using Ellipsoids** Multidimensional scaling [De Leeuw, 1977] aims at embedding points $\mathbf{x}_1, \ldots, \mathbf{x}_n$ in a finite metric space in a lower dimensional one by minimizing the *stress* $\sum_{ij}(\|\mathbf{x}_i - \mathbf{x}_j\| - \|\mathbf{y}_i - \mathbf{y}_j\|)^2$. In our case, this translates to the minimization of $\mathcal{L}_{\text{MDS}}(\mathbf{a}_1, \ldots \mathbf{a}_n, \mathbf{A}_1, \ldots, \mathbf{A}_n) = \sum_{ij}(\|\mathbf{x}_i - \mathbf{x}_j\| - W_2(\mu_{\mathbf{a}_i, \mathbf{A}_i}, \mu_{\mathbf{a}_j, \mathbf{A}_j}))^2$. This objective can be crudely minimized with a simple gradient descent approach operating on factors as advocated in Section 3, as illustrated in a toy example carried out using data from OECD's PISA study[2].

**Word Embeddings** The skipgram model [Mikolov et al., 2013a] computes word embeddings in a vector space by maximizing the log-probability of observing surrounding context words given an input central word. Vilnis and McCallum [2015] extended this approach to *diagonal* Gaussian embeddings using an energy whose overall principles we adopt here, adapted to elliptical distributions with *full* covariance matrices in the 2-Wasserstein space. For every word $w$, we consider an input (as a word) and an ouput (as a context) representation as an elliptical measure, denoted respectively $\underline{\mu_w}$ and $\nu_w$, both parameterized by a location vector and a scale parameter (stored in factor form).

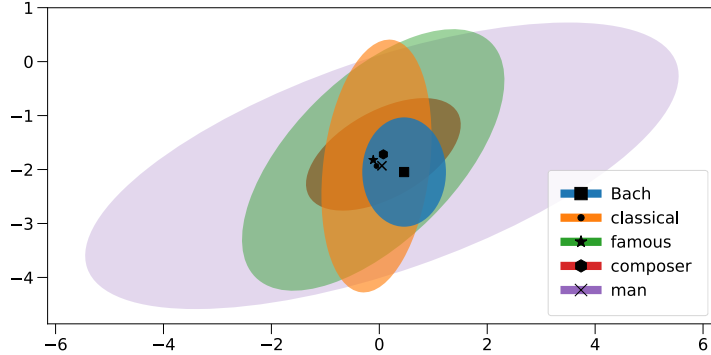

Figure 5: Precision matrix visualization of trained embeddings of a set of words on the plane spanned by the two principal eigenvectors of the covariance matrix of "Bach".

Given a set $\mathcal{R}$ of positive word/context pairs of words $(w, c)$, and for each input word a set $N(w)$ of $n$ negative contexts words sampled randomly, we adapt Vilnis and McCallum's loss function to the $W_2^2$ distance to minimize the following hinge loss:

$$\sum_{(w,c)\in\mathcal{R}} \left[ M - [\mu_w : \nu_c] + \frac{1}{n} \sum_{c'\in N(w)} [\mu_w : \nu_{c'}] \right]_+$$

where $M > 0$ is a margin parameter. We train our embeddings on the concatenated ukWaC and WaCkypedia corpora [Baroni et al., 2009], consisting of about 3 billion tokens, on which we keep only the tokens appearing more than 100 times in the text (for a total number of 261583 different words). We train our embeddings using adagrad [Duchi et al., 2011], sampling one negative context per positive context and, in order to prevent the norms of the embeddings to be too highly correlated with the corresponding word frequencies (see Figure in supplementary material), we use two distinct sets of embeddings for the input and context words.

Table 1: Results for elliptical embeddings (evaluated using our cosine mixture) compared to diagonal Gaussian embeddings trained with the `seomoz` package (evaluated using expected likelihood cosine similarity as recommended by Vilnis and McCallum).

| Dataset | W2G/45/C | Ell/12/CM |
|---|---|---|
| SimLex | **25.09** | 24.09 |
| WordSim | 53.45 | **66.02** |
| WordSim-R | 61.70 | **71.07** |
| WordSim-S | 48.99 | **60.58** |
| MEN | 65.16 | **65.58** |
| MC | 59.48 | **65.95** |
| RG | **69.77** | 65.58 |
| YP | **37.18** | 25.14 |
| MT-287 | **61.72** | 59.53 |
| MT-771 | **57.63** | 56.78 |
| RW | **40.14** | 29.04 |

We compare our full elliptical to diagonal Gaussian embeddings trained using the methods described in [Vilnis and McCallum, 2015] on a collection of similarity datasets by computing the Spearman rank correlation between the similarity scores provided in the data and the scores we compute based on our embeddings. Note that these results are obtained using context ($\nu_w$) rather than input ($\mu_w$) embeddings. For a fair comparison across methods, we set dimensions by ensuring that the number of free parameters remains the same: because of the symmetry in the covariance matrix, elliptical embeddings in dimension $d$ have $d + d(d+1)/2$ free parameters ($d$ for the means, $d(d+1)/2$ for the covariance matrices), as compared with $2d$ for diagonal Gaussians. For elliptical embeddings, we use the common practice of using some form of normalized quantity (a cosine) rather than the direct dot product. We implement this here by computing the mean of two cosine terms, each corresponding separately to mean and covariance contributions:

$$\mathfrak{S}_{\mathfrak{B}}[\mu_{\mathbf{a},\mathbf{A}}, \mu_{\mathbf{b},\mathbf{B}}] := \frac{\langle \mathbf{a}, \mathbf{b} \rangle}{\|\mathbf{a}\|\|\mathbf{b}\|} + \frac{\mathrm{Tr}\,(\mathbf{A}^{\frac{1}{2}}\mathbf{B}\mathbf{A}^{\frac{1}{2}})^{\frac{1}{2}}}{\sqrt{\mathrm{Tr}\mathbf{A}\,\mathrm{Tr}\mathbf{B}}}$$

Using this similarity measure rather than the Wasserstein-Bures dot product is motivated by the fact that the norms of the embeddings show some dependency with word frequencies (see figures in supplementary) and become dominant when comparing words with different frequencies scales. An alternative could have been obtained by normalizing the Wasserstein-Bures dot product in a more standard way that pools together means and covariances. However, as discussed in the supplementary material, this choice makes it harder to deal with the variations in scale of the means and covariances, therefore decreasing performance.

We also evaluate our embeddings on the Entailment dataset ([Baroni et al., 2012]), on which we obtain results roughly comparable to those of [Vilnis and McCallum, 2015]. Note that contrary to the similarity experiments, in this framework using the (unsymmetrical) KL divergence makes sense and possibly gives an advantage, as it is possible to choose the order of the arguments in the KL divergence between the entailing and entailed words.

Table 2: Entailment benchmark: we evaluate our embeddings on the Entailment dataset using average precision (AP) and F1 scores. The threshold for F1 is chosen to be the best at test time.

| Model | AP | F1 |
|---|---|---|
| W2G/45/Cosine | 0.70 | 0.74 |
| W2G/45/KL | 0.72 | 0.74 |
| Ell/12/CM | 0.70 | 0.73 |

**Hypernymy** In this experiment, we use the framework of [Nickel and Kiela, 2017] on hypernymy relationships to test our embeddings. A word A is said to be a *hypernym* of a word B if any B is a type of A, e.g. any *dog* is a type of *mammal*, thus constituting a tree-like structure on nouns. The WORDNET dataset [Miller, 1995] features a transitive closure of 743,241 hypernymy relations on 82,115 distinct nouns, which we consider as an undirected graph of relations $\mathcal{R}$. Similarly to the skipgram model, for each noun $u$ we sample a fixed number $n$ of negative examples and store them in set $\mathcal{N}(u)$ to optimize the following loss: $\sum_{(u,v)\in\mathcal{R}} \log \frac{e^{[\mu_u, \mu_v]}}{e^{[\mu_u, \mu_v]} + \sum_{v'\in\mathcal{N}(u)} e^{[\mu_u, \mu_{v'}]}}$.

We train the model using SGD with only one set of embeddings. The embeddings are then evaluated on a link reconstruction task: we embed the full tree and rank the similarity of each positive hypernym pair $(u, v)$ among all negative pairs $(u, v')$ and compute the mean rank thus achieved as well as the mean average precision (MAP), using the Wasserstein-Bures dot product as the similarity measure. Elliptical embeddings consistently outperform Poincare embeddings for dimensions above a small threshold, as shown in Figure 6, which confirms our intuition that the addition of a notion of variance or uncertainty to point embeddings allows for a richer and more significant representation of words.

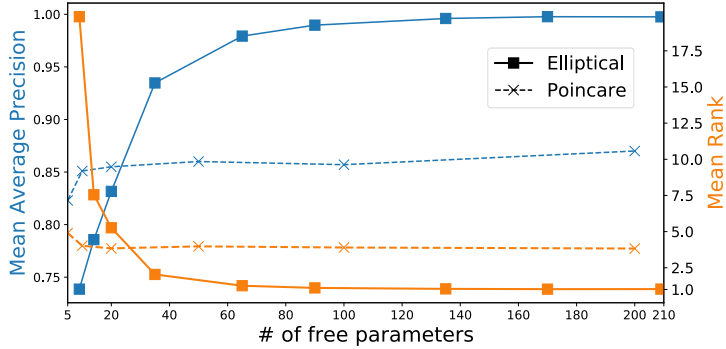

Figure 6: Reconstruction performance of our embeddings against Poincare embeddings (reported from [Nickel and Kiela, 2017], as we were not able to reproduce scores comparable to these values) evaluated by mean retrieved rank (lower=better) and MAP (higher=better).

**Conclusion** We have proposed to use the space of elliptical distributions endowed with the $W_2$ metric to embed complex objects. This latest iteration of probabilistic embeddings, in which a point an object is represented as a probability measure, can consider elliptical measures (including Gaussians) with arbitrary covariance matrices. Using the $W_2$ metric we can provides a natural and seamless generalization of point embeddings in $\mathbb{R}^d$. Each embedding is described with a location $\mathbf{c}$ and a scale $\mathbf{C}$ parameter, the latter being represented in practice using a factor matrix $\mathbf{L}$, where $\mathbf{C}$ is recovered as $\mathbf{L}\mathbf{L}^T$. The visualization part of work is still subject to open questions. One may seek a different method than that proposed here using precision matrices, and ask whether one can include more advanced constraints on these embeddings, such as inclusions or the presence (or absence) of intersections across ellipses. Handling multimodality using mixtures of Gaussians could be pursued. In that case a natural upper bound on the $W_2$ distance can be computed by solving the OT problem between these mixtures of Gaussians using a simpler proxy: consider them as discrete measures putting Dirac masses in the space of Gaussians endowed with the $W_2$ metric as a ground cost, and use the optimal cost of that proxy as an upper bound of their Wasserstein distance. Finally, note that the set of elliptical measures $\mu_{\mathbf{c},\mathbf{C}}$ endowed with the Bures metric can also be interpreted, given that $\mathbf{C} = \mathbf{L}\mathbf{L}^T, \mathbf{L} \in \mathbb{R}^{d\times k}$, and writing $\tilde{\mathbf{l}}_i = \mathbf{l}_i - \bar{\mathbf{l}}$ for the centered column vectors of $\mathbf{L}$, as a discrete point cloud $(\mathbf{c} + \frac{1}{\sqrt{k}}\tilde{\mathbf{l}}_i)_i$ endowed with a $W_2$ metric only looking at their first and second order moments. These $k$ points, whose mean and covariance matrix match $\mathbf{c}$ and $\mathbf{C}$, can therefore fully characterize the geometric properties of the distribution $\mu_{\mathbf{c},\mathbf{C}}$, and may provide a simple form of multimodal embedding.

## Footnotes

[1]For instance, the random variable $Y$ in $\mathbb{R}^2$ obtained by duplicating the same normal random variable $X$ in $\mathbb{R}$, $Y = [X, X]$, is supported on a line in $\mathbb{R}^2$ and has no density w.r.t the Lebesgue measure in $\mathbb{R}^2$.

[2]http://pisadataexplorer.oecd.org/ide/idepisa/

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
