[Supplementary Material]

# Supplementary Material

## Equivalent formulations of $\mathbf{T^{AB}}$

$\mathbf{T^{AB}}$ is defined as the unique PSD matrix verifying $\mathbf{T^{AB}AT^{AB}} = \mathbf{B}$. Using this definition, we derive two equivalent formulations for $\mathbf{T^{AB}}$ :

$$\mathbf{T^{AB}} = \mathbf{A}^{-\frac{1}{2}}(\mathbf{A}^{\frac{1}{2}}\mathbf{B}\mathbf{A}^{\frac{1}{2}})^{\frac{1}{2}}\mathbf{A}^{-\frac{1}{2}}$$
$$= \mathbf{B}^{\frac{1}{2}}(\mathbf{B}^{\frac{1}{2}}\mathbf{A}\mathbf{B}^{\frac{1}{2}})^{-\frac{1}{2}}\mathbf{B}^{\frac{1}{2}}$$

The first is derived as in [Malagò et al., 2018]:

$$\mathbf{T^{AB}AT^{AB}} = \mathbf{B}$$
$$\mathbf{A}^{\frac{1}{2}}\mathbf{T^{AB}}\mathbf{A}^{\frac{1}{2}}\mathbf{A}^{\frac{1}{2}}\mathbf{T^{AB}}\mathbf{A}^{\frac{1}{2}} = \mathbf{A}^{\frac{1}{2}}\mathbf{B}\mathbf{A}^{\frac{1}{2}}$$
$$\mathbf{A}^{\frac{1}{2}}\mathbf{T^{AB}}\mathbf{A}^{\frac{1}{2}} = \left(\mathbf{A}^{\frac{1}{2}}\mathbf{B}\mathbf{A}^{\frac{1}{2}}\right)^{\frac{1}{2}}$$
$$\mathbf{T^{AB}} = \mathbf{A}^{-\frac{1}{2}}\left(\mathbf{A}^{\frac{1}{2}}\mathbf{B}\mathbf{A}^{\frac{1}{2}}\right)^{\frac{1}{2}}\mathbf{A}^{-\frac{1}{2}}$$

We then adapt this derivation to obtain a second formulation of $\mathbf{T^{AB}}$:

$$\mathbf{T^{AB}AT^{AB}} = \mathbf{B}$$
$$\left(\mathbf{T^{AB}}\right)^{-1}\mathbf{B}\left(\mathbf{T^{AB}}\right)^{-1} = \mathbf{A}$$
$$\mathbf{B}^{\frac{1}{2}}\left(\mathbf{T^{AB}}\right)^{-1}\mathbf{B}^{\frac{1}{2}}\mathbf{B}^{\frac{1}{2}}\left(\mathbf{T^{AB}}\right)^{-1}\mathbf{B}^{\frac{1}{2}} = \mathbf{B}^{\frac{1}{2}}\mathbf{A}\mathbf{B}^{\frac{1}{2}}$$
$$\mathbf{B}^{\frac{1}{2}}\left(\mathbf{T^{AB}}\right)^{-1}\mathbf{B}^{\frac{1}{2}} = \left(\mathbf{B}^{\frac{1}{2}}\mathbf{A}\mathbf{B}^{\frac{1}{2}}\right)^{\frac{1}{2}}$$
$$\left(\mathbf{T^{AB}}\right)^{-1} = \mathbf{B}^{-\frac{1}{2}}\left(\mathbf{B}^{\frac{1}{2}}\mathbf{A}\mathbf{B}^{\frac{1}{2}}\right)^{\frac{1}{2}}\mathbf{B}^{-\frac{1}{2}}$$
$$\mathbf{T^{AB}} = \mathbf{B}^{\frac{1}{2}}\left(\mathbf{B}^{\frac{1}{2}}\mathbf{A}\mathbf{B}^{\frac{1}{2}}\right)^{-\frac{1}{2}}\mathbf{B}^{\frac{1}{2}}$$

## Derivation of the Riemannian gradient updates

From [Malagò et al., 2018], we have that the $\exp$ and $\log$ maps of the Riemannian Bures metric are given by:

$$\exp_{\mathbf{C}}(\mathbf{V}) = (\mathcal{L}_{\mathbf{C}}(\mathbf{V}) + \mathbf{I})\,\mathbf{C}\,(\mathcal{L}_{\mathbf{C}}(\mathbf{V}) + \mathbf{I})$$
$$\log_{\mathbf{C}}(\mathbf{B}) = \left(\mathbf{T^{CB}} - \mathbf{I}\right)\mathbf{C} + \mathbf{C}\left(\mathbf{T^{CB}} - \mathbf{I}\right)$$

where $\mathcal{L}_{\mathbf{C}}(\mathbf{V})$ is the solution of *Lyapunov* equation $\mathcal{L}_{\mathbf{C}}(\mathbf{V})\mathbf{C} + \mathbf{C}\mathcal{L}_{\mathbf{C}}(\mathbf{V}) = \mathbf{V}$. One can show that the $\mathcal{L}_{\mathbf{C}}$ operator is linear, and that the following identity holds: $\mathcal{L}_{\mathbf{C}}(\mathbf{XC} + \mathbf{CX})$. In particular, $\mathcal{L}_{\mathbf{C}}(\log_{\mathbf{C}}\mathbf{B}) = \mathbf{T^{CB}} - \mathbf{I}$.

From this, since $\text{grad}_{\mathbf{A}} \frac{1}{2}\mathfrak{B}^2(\mathbf{A}, \mathbf{B}) = -\log_{\mathbf{A}} \mathbf{B}$, the Riemannian gradient update is given by

$$
\begin{aligned}
\mathbf{A}_{t+1} &= \exp_{\mathbf{A}_t}(\eta_t \log_{\mathbf{A}_t} \mathbf{B}) \\
&= \left(\eta_t \mathcal{L}_{\mathbf{A}_t}(\log_{\mathbf{A}_t} \mathbf{B}) + \mathbf{I}\right) \mathbf{A}_t \left(\eta_t \mathcal{L}_{\mathbf{A}_t}(\log_{\mathbf{A}_t} \mathbf{B}) + \mathbf{I}\right) \\
&= \left((1 - \eta_t)\mathbf{I} + \eta_t \mathbf{T}^{\mathbf{A}_t\mathbf{B}}\right) \mathbf{A}_t \left((1 - \eta_t)\mathbf{I} + \eta_t \mathbf{T}^{\mathbf{A}_t\mathbf{B}}\right)
\end{aligned}
$$

## Derivation of the Euclidean gradient

**Notations:** $\otimes$ is the Kronecker product of matrices. Recall that
$$
[\mathbf{B}^\top \otimes \mathbf{A}]\text{vec}(\mathbf{X}) = \text{vec}(\mathbf{A}\mathbf{X}\mathbf{B})
$$
$$
[\mathbf{A} \otimes \mathbf{B}][\mathbf{C} \otimes \mathbf{D}] = [\mathbf{A}\mathbf{C} \otimes \mathbf{B}\mathbf{D}]
$$
In the following, we will often omit the $\text{vec}(.)$ and treat matrices as vectors when the context makes it clear. We will make use of the following identities:
$$
\partial_{\mathbf{X}} f \circ g(\mathbf{X}) = \partial_{\mathbf{X}} f(g(\mathbf{X}))\partial_{\mathbf{X}} g(\mathbf{X})
$$
$$
\partial_{\mathbf{X}} (fg)(\mathbf{X}) = [g(\mathbf{X})^\top \otimes \mathbf{I}]\partial_{\mathbf{X}} f(\mathbf{X}) + [\mathbf{I} \otimes g(\mathbf{X})]\partial_{\mathbf{X}} g(\mathbf{X})
$$
and
$$
\partial_{\mathbf{X}} X^{\frac{1}{2}} = [\mathbf{X}^{\frac{1}{2}} \otimes \mathbf{I} + \mathbf{I} \otimes \mathbf{X}^{\frac{1}{2}}]^{-1}
$$

Let $f(\mathbf{A}, \mathbf{B}) = \text{Tr}(\mathbf{B}^{\frac{1}{2}}\mathbf{A}\mathbf{B}^{\frac{1}{2}})^{\frac{1}{2}}$.

Let us differentiate $f$ w.r.t $\mathbf{A}$ :

$$
\begin{aligned}
\nabla_{\mathbf{A}} f(\mathbf{A}, \mathbf{B}) &= \left[\partial_{\mathbf{A}}(\mathbf{B}^{\frac{1}{2}}\mathbf{A}\mathbf{B}^{\frac{1}{2}})^{\frac{1}{2}}\right]^\top \mathbf{I} \\
&= \left[\left[(\mathbf{B}^{\frac{1}{2}}\mathbf{A}\mathbf{B}^{\frac{1}{2}})^{\frac{1}{2}} \otimes \mathbf{I} + \mathbf{I} \otimes (\mathbf{B}^{\frac{1}{2}}\mathbf{A}\mathbf{B}^{\frac{1}{2}})^{\frac{1}{2}}\right]^{-1} \partial_{\mathbf{A}}(\mathbf{B}^{\frac{1}{2}}\mathbf{A}\mathbf{B}^{\frac{1}{2}})\right]^\top \mathbf{I} \\
&= \left[\mathbf{B}^{\frac{1}{2}} \otimes \mathbf{B}^{\frac{1}{2}}\right]\left[(\mathbf{B}^{\frac{1}{2}}\mathbf{A}\mathbf{B}^{\frac{1}{2}})^{\frac{1}{2}} \otimes \mathbf{I} + \mathbf{I} \otimes (\mathbf{B}^{\frac{1}{2}}\mathbf{A}\mathbf{B}^{\frac{1}{2}})^{\frac{1}{2}}\right]^{-1} \mathbf{I} \\
&= \left[\mathbf{B}^{\frac{1}{2}} \otimes \mathbf{B}^{\frac{1}{2}}\right] \frac{1}{2}(\mathbf{B}^{\frac{1}{2}}\mathbf{A}\mathbf{B}^{\frac{1}{2}})^{-\frac{1}{2}} \\
&= \frac{1}{2}\mathbf{B}^{\frac{1}{2}}(\mathbf{B}^{\frac{1}{2}}\mathbf{A}\mathbf{B}^{\frac{1}{2}})^{-\frac{1}{2}}\mathbf{B}^{\frac{1}{2}}
\end{aligned}
$$

Therefore $\nabla_{\mathbf{A}} f(\mathbf{A}, \mathbf{B}) = \frac{1}{2}\mathbf{T}^{\mathbf{A}\mathbf{B}}$

Let now $\mathbf{A} = \mathbf{L}\mathbf{L}^\top$, let us differentiate w.r.t $\mathbf{L}$ :

$$
\begin{aligned}
\nabla_{\mathbf{L}} f(\mathbf{L}\mathbf{L}^\top, \mathbf{B}) &= \left[\partial_{\mathbf{L}}(\mathbf{B}^{\frac{1}{2}}\mathbf{A}\mathbf{B}^{\frac{1}{2}})^{\frac{1}{2}}\right]^\top \mathbf{I} \\
&= \partial_{\mathbf{L}} \mathbf{A}^\top \left[\partial_{\mathbf{A}}(\mathbf{B}^{\frac{1}{2}}\mathbf{A}\mathbf{B}^{\frac{1}{2}})^{\frac{1}{2}}\right]^\top \mathbf{I} \\
&= \left[\mathbf{L}^\top \otimes \mathbf{I}\right][\mathbf{I} + \mathbf{T}_{n,n}]\frac{1}{2}\mathbf{B}^{\frac{1}{2}}(\mathbf{B}^{\frac{1}{2}}\mathbf{A}\mathbf{B}^{\frac{1}{2}})^{-\frac{1}{2}}\mathbf{B}^{\frac{1}{2}} \\
&= \mathbf{B}^{\frac{1}{2}}(\mathbf{B}^{\frac{1}{2}}\mathbf{A}\mathbf{B}^{\frac{1}{2}})^{-\frac{1}{2}}\mathbf{B}^{\frac{1}{2}}\mathbf{L}
\end{aligned}
$$

where $\mathbf{T}_{n,n}$ is the transposition tensor, such that $\forall \mathbf{X} \in \mathbb{R}^{n \times n}, \mathbf{T}_{n,n}\text{vec}(\mathbf{X}) = \text{vec}(\mathbf{X}^\top)$.

Therefore $\nabla_{\mathbf{L}} f(\mathbf{L}\mathbf{L}^\top, \mathbf{B}) = \mathbf{T}^{\mathbf{A}\mathbf{B}}\mathbf{L}$.

Using the same calculations, one can see that if $\mathbf{A} = \mathbf{L}\mathbf{L}^\top + \varepsilon\mathbf{I}$, then we still have
$$
\nabla_{\mathbf{L}} f(\mathbf{L}\mathbf{L}^\top + \varepsilon\mathbf{I}, \mathbf{B}) = \mathbf{T}^{\mathbf{A}\mathbf{B}}\mathbf{L}
$$
since $\partial_{\mathbf{L}} \left[\mathbf{L}\mathbf{L}^\top + \varepsilon\mathbf{I}\right] = \partial_{\mathbf{L}} \left[\mathbf{L}\mathbf{L}^\top\right]$

## Model Hyperparameters and Training Details

**Word Embeddings**   We train our embeddings on the concatenated ukWaC and WaCkypedia corpora [Baroni et al., 2009], consisting of about 3 billion tokens, on which we keep only the tokens appearing more than 100 times in the text after lowercasing and removal of all punctuation (for a total number of 261583 different words). We optimize 5 epochs using adagrad [Duchi et al., 2011] with $\epsilon = 10^{-8}$ with a learning rate of $0.01$. We use a window size of 10 (i.e. positive examples consist of the first 5 preceding and first 5 succeeding words), set the margin to $10$, sample one negative context per positive context and, in order to prevent the norms of the embeddings to be too highly correlated with the corresponding word frequencies (see Figure 7), we use two distinct sets of embeddings for the input and context words. In order to use as much parallelization as possible, we use batches of size 10000, but believe that smaller batches would lead to improved performances. We limit matrix square root approximations to 6 Newton-Schulz iterations and add $0.01\mathbf{I}$ to the covariances to ensure non-singularity.

To generate batches, we use the same sampling tricks as in [Mikolov et al., 2013b], namely sub-sampling the frequent terms (using a threshold of $10^{-5}$ as recommended for large datasets) and smoothing the negative distribution by using probabilities $\{f_i^{3/4}/Z\}$ where $f_i$ is the frequency of word $i$ for sampling negative contexts $\{c_i'\}$.

We then evaluate our embeddings on the following datasets: Simlex [Hill et al., 2015], WordSim [Finkelstein et al., 2002], MEN [Bruni et al., 2014], MC [Miller and Charles, 1991], RG [Rubenstein and Goodenough, 1965], YP [Yang and Powers, 2005], MTurk [Radinsky et al., 2011] [Halawi et al., 2012], RW [thang Luong et al., 2013], using the context embeddings and the Wasserstein-Bures cosine as a similarity measure.

**Hypernymy**   We train our embeddings on the transitive closure of the `WORDNET` dataset [Miller, 1995] which features 743,241 hypernymy relations on 82,115 distinct nouns. For disambiguation, note that if $(u, v)$ is a hypernymy relation with $u \neq v$, $(v, u)$ is in general *not* a positive relation, but $(u, u)$ is as a noun is always its own hypernym.

We perform our optimization using SGD with batches of 1000 relations, a learning rate 0.02 for dimensions 3 and 4 and 0.01 for higher dimensions, sample 50 negative examples per positive relation, use 6 square root iterations and add $0.01\mathbf{I}$ to the covariances. Contrary to the skipgram experiment, we use a single set of embeddings and use the Wasserstein-Bures dot product as a similarity measure.

## Wasserstein-Bures Cosine

(a) inputs                                                    (b) contexts

Figure 7: log-log plot of the traces of the embeddings' covariances vs. word frequency: the sizes of the input embeddings follow a power law, whereas context embeddings give less importance to very frequent words and emphasize on medium frequency words.

(a) inputs                                  (b) contexts

Figure 8: log-log plot of the norms of the embeddings' means vs. word frequency: the sizes of the input embeddings follow a power law, whereas context embeddings give less importance to very frequent words and emphasize on medium frequency words.

As discussed in section 4, a natural choice of similarity measure would be the Wasserstein-Bures cosine, obtained by normalizing the Wasserstein-Bures dot product with the means' norms and covariances' root traces jointly:

$$\cos_{\mathfrak{B}}[\rho_{\mathbf{a},\mathbf{A}}, \rho_{\mathbf{b},\mathbf{B}}] := \frac{\langle \mathbf{a}, \mathbf{b}\rangle + \mathrm{Tr}[\mathbf{A}^{\frac{1}{2}}\mathbf{B}\mathbf{A}^{\frac{1}{2}}]^{\frac{1}{2}}}{(\|\mathbf{a}\|^2 + \mathrm{Tr}\mathbf{A})^{\frac{1}{2}}(\|\mathbf{b}\|^2 + \mathrm{Tr}\mathbf{B})^{\frac{1}{2}}}$$

However, we have found that in some applications (and notably in our skipgram experiments) such a joint normalization can result in either the means or the covariances to have a negligible contribution if the scales of the parameters differ too much. To circumvent this problem, we introduce another similarity measure, which is a mixture of two cosine terms:

$$\mathfrak{S}_{\mathfrak{B}}[\rho_{\mathbf{a},\mathbf{A}}, \rho_{\mathbf{b},\mathbf{B}}] := \frac{\langle \mathbf{a}, \mathbf{b}\rangle}{\|\mathbf{a}\|\|\mathbf{b}\|} + \frac{\mathrm{Tr}[\mathbf{A}^{\frac{1}{2}}\mathbf{B}\mathbf{A}^{\frac{1}{2}}]^{\frac{1}{2}}}{\sqrt{\mathrm{Tr}\mathbf{A}\,\mathrm{Tr}\mathbf{B}}}$$

This latter similarity measure allows to gather information from the means and the covariances independently. Note that while the term corresponding to the covariances is obtained in a cosine-like normalization, it takes values between 0 and 1 as it only involve traces of PSD matrices, whereas the means term is a regular Euclidean cosine and therefore takes values between -1 and 1. We compare the behaviors of these two measures on the word similarity evaluation task by introducing a mixing coefficient $\rho$, and defining

$$\cos_{\mathfrak{B}}[\rho_{\mathbf{a},\mathbf{A}}, \rho_{\mathbf{b},\mathbf{B}}; \rho] := \frac{\langle \mathbf{a}, \mathbf{b}\rangle + \rho\mathrm{Tr}[\mathbf{A}^{\frac{1}{2}}\mathbf{B}\mathbf{A}^{\frac{1}{2}}]^{\frac{1}{2}}}{(\|\mathbf{a}\|^2 + \rho\mathrm{Tr}\mathbf{A})^{\frac{1}{2}}(\|\mathbf{b}\|^2 + \rho\mathrm{Tr}\mathbf{B})^{\frac{1}{2}}}$$

$$\mathfrak{S}_{\mathfrak{B}}[\rho_{\mathbf{a},\mathbf{A}}, \rho_{\mathbf{b},\mathbf{B}}; \rho] := \frac{\langle \mathbf{a}, \mathbf{b}\rangle}{\|\mathbf{a}\|\|\mathbf{b}\|} + \rho\frac{\mathrm{Tr}[\mathbf{A}^{\frac{1}{2}}\mathbf{B}\mathbf{A}^{\frac{1}{2}}]^{\frac{1}{2}}}{\sqrt{\mathrm{Tr}\mathbf{A}\,\mathrm{Tr}\mathbf{B}}}$$

As can be seen from figure 9, the Wasserstein-Bures cosine is less well behaved and makes it difficult to find an optimal mixing value. On the other hand, the mixture of cosines similarity measure varies more smoothly and seems to reach a performance maximum around $\rho = 1$, and achieves better performance than the Wasserstein-Bures cosine on most datasets.

(a) $\mathfrak{S}_{\mathfrak{B}}$

(b) $\cos_{\mathfrak{B}}$

Figure 9: Pearson rank correlation scores on similarity benchmarks as a function of the mixing coefficient: $\mathfrak{S}_{\mathfrak{B}}$ smoothly attains a maximum in performance around $\rho = 1$, whereas $\cos_{\mathfrak{B}}$ has a not so smooth behavior.