[Reviews · NeurIPS 2018]

Reviewer 1



The paper proposes using elliptical distribution (specifically uniform elliptical distributions) as probability measures for representing data instances in an embedding space. The work is inspired by prior work where the KL divergence was used for Gaussian distribution based embeddings. A motivating factor versus previous work is ensuring consistent behavior as the distributions collapse to Dirac delta functions. The papers use the 2-Wasserstein divergence/distance (motivated by its connection to optimal transport) as the metric for the embedding space. This divergence/distance has a known closed form expression for uniform elliptical distributions that relies on the Bures/Kakutani distance metric. The paper then explores manifold optimizations of the divergence in terms of the covariance matrix, and uses a factored parameterization of the covariance matrix for the optimization. To compute the gradient of the Bures distance a Newton-Schulz algorithm is used to find the matrix square roots and their inverses, which is a more parallelizable manner than the eigendecompostion. The approach is applied to a simple multidimensional scaling example and then to word embedding data sets that have previously been used with other probability distribution based embeddings. Strength: A novel idea that is fairly well conveyed. The discussion of the manifold optimization and Figure 2 are useful. The optimization of the gradient implementation seems to be well designed. It seems like a fairly complete piece of work with the potential for further refinement and application. Weaknesses: The text should include more direct motivation on why the added flexibility of an elliptical embedding is useful for modeling versus standard point embedding, especially relating to the hypernymy experiment. The equation between lines 189 and 190 appears to be wrong. By Equation 2, the trace term should be divided by $d+2$, unless the covariances are pre-scaled. The arguments (including Figure 3) for using the precision versus the covariance for visualization could be removed. The counterintuitive nature appears to be subjective and distracts from the modeling purpose of a probability distribution based embedding. Reference for the tractable (computable) form of Bures distance should be Uhlmann. A. Uhlmann, "The “transition probability” in the state space of a∗-algebra," Rep. Math. Phys. 9, 273 (1976) Some issues with typesetting appear to be problems with inline figures and sectioning. Minor errors and corrections: Line 25 "are extremely small (2 or 3 numbers" -> "can be represented in two or three dimensional space. Line 37 add comma around the phrase "or more ... $\langle y_i,y_j\rangle$" Line 55 The phrase "test time" is not clear in this context. Line 55 I cannot find any relevant comment in Table 1. Line 63 "scenarii" Line 69 Should notation for $M$ in pseudo-inverse match other cases (i.e., boldface)? Line 117 Typesetting and line number are messed up by inline figure 1. Line 130 $B$ should be $b$, and $C$ should be $B$. Line 132 Reasoning for the counterintuitive visualizations is that for uniform distributions the divergence decreases if the scale decreases since the density is more concentrated. Line 146 "adressed" Line 152 Here and elsewhere the references appears sometimes with both parentheses and brackets. Line 155 "on the contrary" to what? This seems to be incorrectly copied from the introductory paragraph in Section 3. Line 157 "to expensive" Line 166 "w.r.t" Line 177 Should be supplementary material or supplement. Line 183.1 Should be "a general loss function" since $\matchal{L}$ is indicated. Line 214 "matrics" Line 215 "would require to perform" "would require one to perform" Line 233 Elliptical should be lower case Line 236 "is to use their precision matrix their covariances" ? Line 242 $y_i,y_j$ -> $y_i-y_j$ Line 243 Missing comma before $\mathbf{a}_n$ Figure 5 caption fix the initial quotation mark before Bach. Line 289 "asymetry" Line 310 "embedd" Line 315 Should clarify if this is the Bures dot product or the Bures cosine. Line 323 "interesections" The reference list is poorly done: lack of capitalization of proper nouns, inconsistent referencing to journal names (some with abbreviations some without, some missing title case), some URL s and ISSN inserted sporadically, and Inconsistent referencing to articles on arXiv. ------ Author rebuttal response ----- The author rebuttal helped clarify and I have raised my score and edited the above. The authors have promised to clarify the dot product equation that appears inconsistent to me. I agree with Reviewer 3 that the visualization issue is not convincing and would caution elaborating on this somewhat tangential issue more. The original intuition was that the distance between a point and a uniform ellipsoid increase as the scale of the latter increases. However, with the additive nature of the similarity metric (dot product) used for the word embedding this doesn't appear to be the case. Switching the covariance with precision for the uniform elliptical embeddings seems to make it more confusing for Figure 5, where the embedding is trained using the dot product rather than distance. I also hope the authors take care in carefully proofreading the manuscript and adhering to NIPS template for typesetting.

Reviewer 2



Response to feedback: Thanks for your feedback and especially for clarifying issues around reproducibility and details concerning the hypernymy task and comparison with the N-K reference. Many of my questions stemmed from having looked at their arxiv version and not the final NIPS version which indeed has some of the clarifications you mentioned in the rebuttal. It is reassuring that the code for your experiments will be made available on-line. On the negative side, the rebuttal failed to address my concern that some discussion should have been provided in connection with the most compelling set of experimental around the hypernymy task, about the source of these gains and how they might be grounded in properties of the proposed elliptical embeddings. Are these gains also possible with the Vilnis-McCullum approach? Or will we see similar inconclusive comparison as in Table 2? Main ideas: The authors study embedding into the 2-Wasserstein metric space of elliptical probability distributions whose pdfs have ellipsoidal level sets. The simplest case and of main focus in the paper is that of uniform distributions over ellipsoids. Such distributions are characterized by a mean vector and covariance matrix and the 2-Wasserstein distance between any pair can be expressed in closed form in terms of these parameters, as established in the literature. Also recalled from the literature is the Bures product which is a counterpart of the Euclidean inner product in these spaces. A factorized representation of the covariance matrix and the corresponding steps to efficiently carry out SGD (wrt to this parameterization) on functionals of pairwise embedding distances and/or Bures products is described. Embeddings are computed for several different applications involving multidimensional scaling, word embeddings, entailment embeddings, and hypernymy where the latter refers to 'type of' relationships among nouns, as conveyed by a directed acyclic graph WORDNET. Relation to prior work: Prior work of Vilnis and McCallum proposed embedding into diagonal Gaussians with KL divergence metric. It is claimed that proposed approach avoids some drawbacks of KL divergence such as not blowing up as Gaussian variances decrease and a graceful reduction to vector embedding with Euclidean metric in such cases. Strengths: 1. Excellent presentation of all the necessarily tools from the literature that are required for optimizing embeddings, from the closed form Wasserstein distances, to the algorithm for computing matrix square roots. 2. Nice complexity reduction trick via (7). 3. Significantly improved experimental results for Hypernymy modeling over prior work, as shown in Figure 6. Weaknesses: 1. Concerning K-L divergence, though it blows up with decreasing variance, it does behave like Euclidean distance between means scaled by the inverse of the variance. 2. Insufficient details for reproducing word embeddings experiments such as the data set used for training, parameter initialization, context window size, number epochs over data set, etc. 3. The experimental results on word embeddings in Tables 1 and 2 are not very compelling as compared to Gaussian embeddings, the chief baseline from the literature. 4. For the Hypernymy experiments some confirmation that negative examples (for training and eval) for a relation (u,v) include (v,u) and (u,u) would be reassure that an apples-to-apples comparison is being made wrt. Nickel and Kiela 2017. Also missing are implementation details like how parameters are initialized, number of negatives per positive, training epochs, etc. Also no explanation of choice behind have one embedding for context and target and RMSProp vs Adagrad in the word embedding experiments. 5. Given the significant improvement over Nickel-Kiela Hypernymy results, there should be some attempt to explain the source of these improvements and tie them back to properties of the proposed embedding. This is critical especially given that the Bures product is symmetric while Hypernym relations are not. That the Mean Rank drops to 1 is especially surprising given this. 6. Some discussion about difference between (6) in Nicekl Kiela and l. 303 loss would be nice. In N-K (6), the positive example contribution does not appear in the denominator whereas in l. 303 it does. Typos: l. 18 'entanglement' -> 'entailment' (I think) l. 155: repetition of text starting on line 146 l. 310: 'embedd' -> 'embed' l. 323: 'interesections' -> 'intersections'

Reviewer 3



The paper proposes a method to embed objects into uniform elliptical probability distributions. It, then, proposes to use the Wasserstein distance in the embedding space to compute the discrepancy between distributions, which is advantageous to alternatives such as KL divergence, but is heavy to compute. Selecting Gaussians as target distribution solves the computational burden by offering a closed form solution for this setting. The model is used for learning word representation and in a visualization scenario. - I find this an interesting paper. Many nontrivial technical observations are highlighted and fit well together. This is a nice continuation of the works on embedding to distributions (as opposed to points) and also a natural and good application of Wasserstein distances. Perhaps will bring interest to the paper as a reasult. - To my knowledge the method proposed is novel, and seems appropriate for NIPS. - Figure 1 is nice, informative and useful. - The method is kind of limited, in that, it works only for embedding into a specific target distribution, but is still a good start for this direction of research. - While embedding to "uniform" elliptical distributions is the one figured out in the paper, the title of the paper does not reflect that, which a little looks like an oversell. - The equation after "minimizing the stress" in line 242 has a typo. - The related work is covered well overall. However, a related work which worths mentioning is [1] below, in which object classes are embedded into high dimensional Euclidean norm balls, with parametrized center and radii. The geometry enables for logical constraints such as inclusion and exclusions between classes to be imposed (like in a Venn diagram) and is guaranteed at test time. The relevance of [1] is two-fold, first the embedding is performed into a uniform elliptical. Second, the future works in the current submission mentions that the authors are considering imposing similar logical constraints, which is addressed there. - The experimental results are acceptable. However, the visualization issue which is brought up is not developed enough and arguments are not convincing. The paper is well structured and overall well written. However, the text write up seems rushed, with a large number of spelling problems in the text. - minor: "the interest of" -> the interestingness of? "entanglement" -> entailment "matrix their covariances" "gaussian" "euclidean: "embedd" "asymetry" "sparameter" "prove to expensive" Overall, I vote for the paper to be accepted after proofreading. ----------- [1] F. Mirzazadeh, S. Ravanbakhsh, N. Ding, D. Schuurmans, "Embedding inference for structured multilabel prediction", In NIPS 2015. ------------------------------------- After reading other reviews, author rebuttal, and reviewer discussions, my remaining concerns are: - The visualization arguments can be improved. - Citation to [1] above is missing. - Intuitive discussion about why the method proposed should have advantages over the main competitors would be useful. The rest of concerns raised in my review is addressed in the rebuttal.